# A Case of Non-Syndromic Congenital Cataracts Caused by a Novel *MAF* Variant in the C-Terminal DNA-Binding Domain—Case Report and Literature Review

**DOI:** 10.3390/genes15060686

**Published:** 2024-05-25

**Authors:** Sharon H. Zhao, Kai Lee Yap, Valerie Allegretti, Andy Drackley, Alexander Ing, Adam Gordon, Andrew Skol, Patrick McMullen, Brenda L. Bohnsack, Sudhi P. Kurup, Hantamalala Ralay Ranaivo, Jennifer L. Rossen

**Affiliations:** 1Department of Ophthalmology, Northwestern University Feinberg School of Medicine, Chicago, IL 60611, USA; sharon.zhao@northwestern.edu (S.H.Z.); bbohnsack@luriechildrens.org (B.L.B.);; 2Department of Pathology and Laboratory Medicine, Ann & Robert H. Lurie Children’s Hospital of Chicago, Chicago, IL 60611, USA; klyap@luriechildrens.org (K.L.Y.); adrackley@luriechildrens.org (A.D.); aing@luriechildrens.org (A.I.); askol@luriechildrens.org (A.S.); patrick.mcmullen@northwestern.edu (P.M.); 3Department of Pathology, Northwestern University Feinberg School of Medicine, Chicago, IL 60611, USA; 4Division of Ophthalmology, Ann & Robert H. Lurie Children’s Hospital of Chicago, Chicago, IL 60611, USA; vallegretti@luriechildrens.org (V.A.); adam.gordon@northwestern.edu (A.G.); hralay@luriechildrens.org (H.R.R.)

**Keywords:** *MAF*, non-syndromic cataract, bilateral congenital cataract, microcornea, whole exome sequencing

## Abstract

The *MAF* gene encodes a transcription factor in which pathogenic variants have been associated with both isolated and syndromic congenital cataracts. We aim to review the *MAF* variants in the C-terminal DNA-binding domain associated with non-syndromic congenital cataracts and describe a patient with a novel, disease-causing de novo missense variant. Published reports of C-terminal *MAF* variants and their associated congenital cataracts and ophthalmic findings were reviewed. The patient we present and his biological parents had genetic testing via a targeted gene panel followed by trio-based whole exome sequencing. A 4-year-old patient with a history of bilateral nuclear and cortical cataracts was found to have a novel, likely pathogenic de novo variant in *MAF*, NM_005360.5:c.922A>G (p.Lys308Glu). No syndromic findings or anterior segment abnormalities were identified. We report the novel missense variant, c.922A>G (p.Lys308Glu), in the C-terminal DNA-binding domain of *MAF* classified as likely pathogenic and associated with non-syndromic bilateral congenital cataracts.

## 1. Introduction

Congenital cataract is a crystalline lens opacification noted within the first year of life and may be an isolated finding or a part of a syndrome. Its etiologies include genetic causes, prenatal infections, and intrauterine exposures [1]. The most commonly identified genetic causes include pathogenic variants in lens crystallins (45%), gap junction connexins (16%), and developmental transcription factors such as *MAF* (MIM #177075) and PITX3 (MIM #602669) (12%) [2]. Further, there is an interplay with environmental factors as evidenced by phenotypic variation among individuals with identical pathogenic variants [3].

We review cases of C-terminal domain variants in *MAF* that have been reported in the literature and summarize the association with non-syndromic congenital cataract. We also describe a 4-year-old patient with bilateral congenital cataracts identified at 6 weeks of age who was found to have a novel variant in the *MAF* gene [4]. 

## 2. Materials and Methods

### 2.1. Patient Information and History

Our patient was referred to the Division of Ophthalmology at the Ann & Robert H. Lurie Children’s Hospital of Chicago at 6 weeks old for bilateral visually significant nuclear and cortical congenital cataracts. The pregnancy was complicated by gestational diabetes and maternal hypothyroidism, and delivery was performed at full term without complications. There was no family history of childhood eye disease or strabismus including no family member (including parents) with congenital cataracts. Family history is only significant for his mother who had glaucoma. Written informed consent for a prospective research protocol was obtained from the parents under an IRB-approved study (IRB 2021-4730), and the study abided the tenets of the Declaration of Helsinki and was conducted in accordance with the Health Insurance Portability and Accountability Act.

The patient underwent sequential lensectomy and anterior vitrectomy of the right eye at 7 and left eye at 8 weeks of age and was subsequently fitted with aphakic contact lenses. His right eye underwent two additional capsulectomy surgeries for capsular phimosis and developed glaucoma following cataract surgery controlled with topical anti-hypertensives at 9 months of age. At that time, axial length was 20.61 mm in the right eye (normal range) and 17.86 mm in the left eye (hyperopic). As a result, his right eye developed anisometropic amblyopia requiring part-time occlusion therapy, and he also became esotropic. He underwent bilateral medial rectus recessions at 4 years of age. At final follow-up at 5 years of age, the patient’s best corrected visual acuity was 20/125 in the right eye and 20/40 in the left eye. The patient remained otherwise healthy and met all developmental milestones. No other anterior segment abnormalities were noted on examination.

### 2.2. Genetic Testing 

The proband underwent pediatric assessments and genetic testing beginning with a comprehensive early-onset cataract gene panel by next-generation sequencing. The laboratory utilized their proprietary bioinformatic analysis pipeline to filter and analyze the 153 targeted genes with NGS reads aligned to genome build GRCh37/hg19; they cited a greater than 99% coverage at a minimum depth of 50×, with copy number variant (CNV) resolution at a single exon for nearly all targeted exons. This testing, initially considered nondiagnostic, was followed by trio-based whole exome sequencing (WES) at a second commercial laboratory to assess for another molecular etiology. Their proprietary analysis pipeline was utilized with reads also aligned to genome build GRCh37/hg19, citing that 98.9% of targeted regions were covered at a minimum depth of 10×, an overall average depth of 143×, and CNV resolution at a level of approximately three or more exons. Finally, the trio WES dataset was re-analyzed by the Lurie Children’s Molecular Diagnostics Laboratory on a research basis, using the Illumina DRAGEN Bio-IT Platform 3.9 with alignment of the data to genome build GRCh38/hg38.

## 3. Results

### 3.1. Exome Sequencing

The initial commercial genetic testing via an early-onset cataract next-generation sequencing gene panel identified a heterozygous variant in *MAF*, NM_005360.5 c.922A>G (p.Lys308Glu), that was classified as a variant of uncertain significance (VUS). Subsequent testing by trio-based WES at a second CLIA-certified laboratory revealed that the aforementioned heterozygous *MAF* variant was not present in either parent and was thus de novo in our patient; the variant was classified and reported as likely pathogenic (LP). Though the variant did not undergo orthogonal confirmation by either laboratory due to satisfaction of their internal quality metrics, its identification by both laboratories utilizing different target enrichment/hybrid capture kits and bioinformation pipelines functions as the orthogonal confirmation of this variant’s presence in our patient. A 3D reconstruction of the protein with the variant is included in Figure 1A (AlphaFold, Google DeepMind) along with *MAF* genetic testing results in our patient and biologic parents in Integrative Genomics Viewer (IGV) in Figure 1B. 

Re-analysis of the trio WES data by the Lurie Children’s Molecular Diagnostics Laboratory did not reveal any additional potentially clinically relevant variants and supported the disease-causing nature of this variant. In addition to this finding being consistent with our patient’s phenotype including congenital cataracts, myopia, and secondary glaucoma, this previously unreported variant is absent from gnomAD [5], is strongly predicted in silico (REVEL score = 0.934) to be damaging to the encoded MAF transcription factor’s structure and/or function [6], and is located within a functionally important domain with little benign variation [7,8]. ACMG-AMP criteria applied are as follows: PS2, PM1, PM2, and PP3 [9].

### 3.2. Literature Review 

A literature review of pathogenic and likely pathogenic variants in the C-terminal region of *MAF* was conducted by HGMD database extraction to evaluate for any genotype–phenotype associations (Table 1) [7,10,11,12,13,14,15,16,17,18,19,20,21,22,23]. Seventeen variants have been described in the literature associated with isolated, non-syndromic congenital cataracts (Table 1, Figure 2), similar to our patient. All variants were missense, as in our patient, and either inherited in an autosomal dominant fashion or de novo (similar to our patient) [7,8,10,11,12,13,14,15,16,17,18,19,20,21,22,23]. Additional ocular findings included microcornea, which was noted in association with one variant in the extended homology region, with four variants in the basic motif, and with one variant in the bZIP domain [7,11,14,16,18,23]. Iris coloboma was also noted in some cases with microcornea but was not present in isolation in any case [7,14,18]. Secondary glaucoma was noted in two patients as seen in our patient. Additionally, intra-familial variable expressivity of different anterior segment manifestations has been observed with pathogenic variants [7,14]. Table 2 is included as a brief summary of experiments using *MAF* mutations in cellular and animal models and their main findings that establish the relationship between the *MAF* domain and the congenital cataract phenotype.

This figure demonstrates that *MAF* variants described in the literature along the C-terminal domain are located in different domains. The variant identified in our patient is flanking the leucine-zipper region.

## 4. Discussion

*MAF*, v-maf musculoaponeurotic fibrosarcoma oncogene homolog, encodes a transcription factor that uses the basic region domain to bind target genes before dimerizing using the leucine zipper domain [29]. Its interactions are stabilized by the extended homology region and result in a conformational change [30]. *MAF* is expressed in the lens placode, vesicle, and fiber proteins and is key to ocular development by regulating lens fiber cell development and chondrocyte terminal differentiation and by increasing the apoptosis susceptibility of T-cells [14,31]. 

Known pathogenic variants in *MAF* are linked to congenital cataracts and anterior segment abnormalities, including microcornea and iris coloboma. The *MAF* protein has two functional domains, with the associated phenotypes dependent on the domain in which the variant is located; Aymé-Gripp syndrome is associated with N-terminal transactivation domain variants, while non-syndromic ocular abnormalities are associated with variants in the C-terminal DNA-binding domain [7,8,12,18,32,33,34]. Aymé-Gripp syndrome is characterized by sensorineural hearing loss, seizures, distinct craniofacial features, short stature, and developmental delay in addition to congenital cataracts [33,34]. In contrast, *MAF* pathogenic variants in the C-terminal domain are associated with a non-syndromic ocular phenotype of childhood cataracts and iris coloboma and/or microcornea [7,10,11,12,13,14,15,16,17,18,19,20,21,22,23]. 

The basic region of the C-terminal is the most frequently impacted region among the known *MAF* variants [7,10,11,12,13,14,15,16,17,18,19,20,21,22,23]. The putative mechanism causing congenital cataracts is the impaired activation of β-crystallin genes and the many non-crystallin genes involved in lens development [35]. Our patient’s variant is in the basic leucine zipper (bZIP) domain within the C-terminal, near other variants associated with congenital cataracts with and without anterior segment abnormalities (UniProt) [36]. While our patient did not have microcornea or coloboma, he did have secondary glaucoma, which has been previously reported in association with two other C-terminal variants, c.809C>A (p.Ser270Tyr) and c.915C>T p.(Cys305Trp) [11,13]. 

Identification of this previously unreported *MAF* variant, c.922A>G (p.Lys308Glu), supports the body of evidence that the C-terminal domain is necessary for appropriate lens development as abnormalities have a causative association with non-syndromic, congenital, and juvenile-onset cataracts with and without anterior segment abnormalities. Importantly, it expands the library of *MAF* variants affecting the C-terminal with variable ocular manifestations. The identification and detailed phenotyping of additional individuals and families with *MAF* C-terminal domain variant will help facilitate the further assessment of genotype–phenotype correlations for this gene–disease relationship. 

## 5. Conclusions

The phenotypic variation seen in *MAF* variants leading to congenital cataracts emphasizes the need for further study of the protein’s role in development and how variants in the C-terminal domain of the gene result in ophthalmologic abnormalities. This understanding could help identify future targeted treatments in individuals with pathogenic *MAF* variants.

In conclusion, our case and review of the pre-existing literature reinforces the association between missense variation in the C-terminal DNA-binding domain of *MAF* and non-syndromic ocular abnormalities, specifically bilateral congenital cataracts. This report serves to foster improved interpretation and classification for *MAF* variants and to strengthen the association between C-terminal *MAF* missense variants and isolated congenital cataracts with additional ocular features.

## Figures and Tables

**Figure 1 genes-15-00686-f001:**
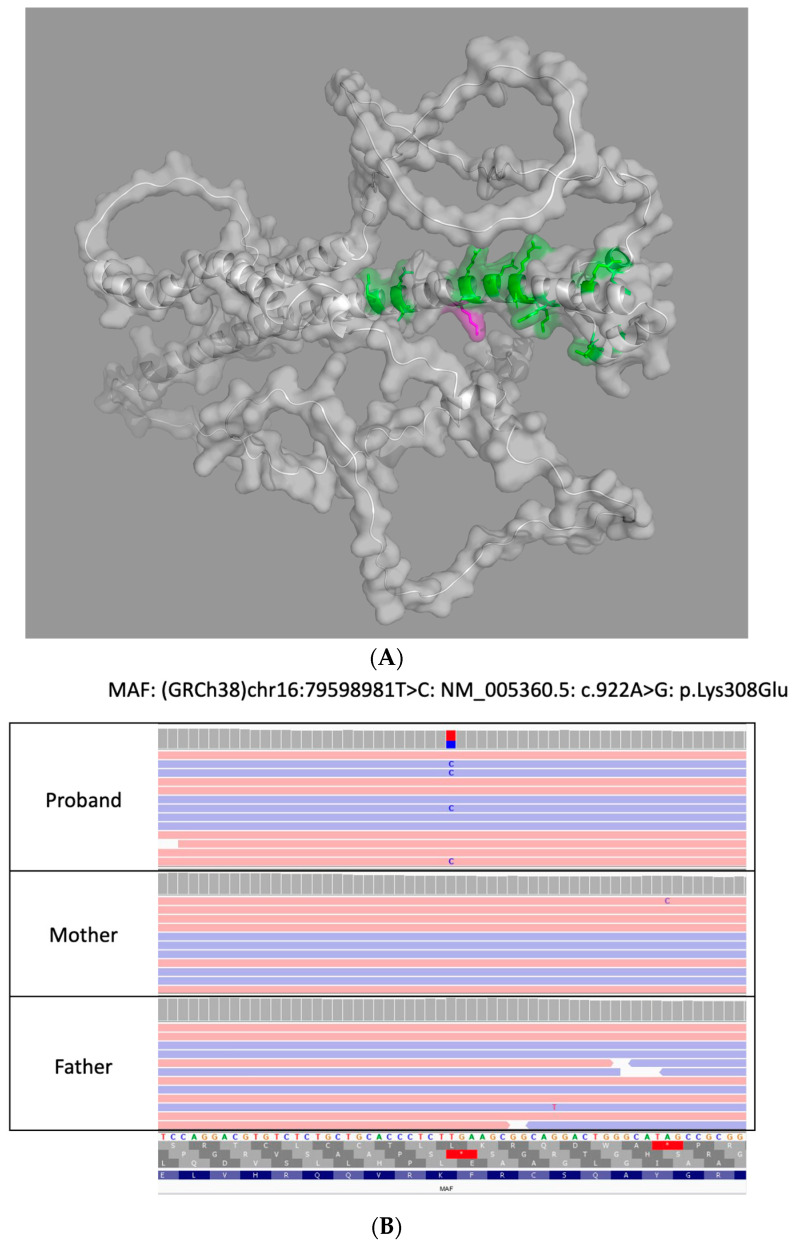
(**A**) 3D Reconstruction of MAF protein in our patient. This is a 3D reconstruction of the altered MAF protein due to our variant (*MAF*, c.922A>G (p.Lys308Glu)) flanking the leucine-zipper region utilizing AlphaFold. (**B**) Genetic testing results of patient and biologic parents. Genetic testing results of patient and biologic parents in Integrative Genomics Viewer (IGV) that highlights the de novo variant in *MAF* identified in our patient with congenital cataracts.

**Figure 2 genes-15-00686-f002:**
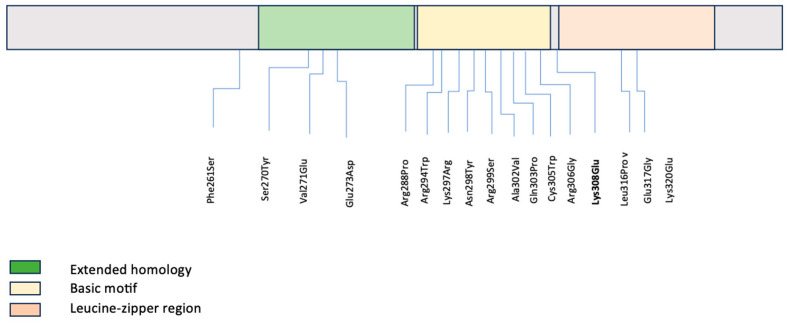
Summary diagram of *MAF* functional domains in the C-terminal including both our novel variant and previously reported variants.

**Table 1 genes-15-00686-t001:** Reported functional variants in the C-terminal of the *MAF* gene associated with non-syndromic congenital cataracts [7,10,11,12,13,14,15,16,17,18,19,20,21,22,23].

Original Report of *MAF* Variant	cDNA Variant	Protein Change	Protein Domain	Variant Type	Cataract Phenotype	Anterior Segment Manifestations	Secondary Glaucoma	Inheritance	ACMG Classification	Citations
Jackson 2020	c.782T>C	Phe261Ser	N/A	Missense	N/A	N/A	No	AD	N/A	[10]
Dudakova 2017	c.809C>A	Ser270Tyr	Extended homology region	Missense	Nuclear cataract	Bilateral microcornea	Yes	AD	P	[11]
Si 2019	c.812T>A	Val271Glu	Extended homology region	Missense	Nuclear cataracts with lamellar opacities	N/A	No	AD	P	[12]
Ma 2016	c.819G>C	Glu273Asp	Extended homology region	Missense	N/A	N/A	No	Presumed de novo or sporadic	LP	[13]
Jamieson 2002	c.863G>C	Arg288Pro	Basic motif	Missense	Family 1—cortical pulverulent cataract with anterior + posterior sutural densities	Family 1—opaque corneas, Peters anomaly	No	AD;	N/A	[14]
					Family 2—cortical, pulverulent, lamellar lens opacities; nuclear cataract	Family 2—two individuals with microcornea, one with bilateral iris coloboma		Family 1 with unbalanced and balanced forms of translocation		
Sun 2014	c.880C>T	Arg294Trp	Basic motif	Missense	Nuclear cataracts	N/A	No	AD	P	[15]
Ma 2016	c.880C>T	Arg294Trp	Basic motif	Missense	N/A	N/A	No	Presumed de novo or sporadic	LP	[13]
Vanita 2006	c.890A>G	Lys297Arg	Basic motif	Missense	Cerulean cataract	Microcornea	No	AD	P	[16]
Patel 2019	c.892A>T	Asn298Tyr	Basic motif	Missense	N/A	N/A	No	Unknown	N/A	[17]
Hansen 2007	c.895C>A	Arg299Ser	Basic motif	Missense	Lamellar (zonular) and star-shaped	Microcornea, occasional iris coloboma	No	AD	N/A	[18]
Rechsteiner 2021	c.905C>T	Ala302Val	Basic motif	Missense	N/A	N/A	No	N/A	LP	[19]
Narumi 2014	c.908A>C	Gln303Pro	Basic motif	Missense	Lamellar cataract	Microcornea and/or iris coloboma in some affected individuals	No	AD	N/A	[7]
Ma 2016	c.915C>T	Cys305Trp	Basic motif	Missense	N/A	N/A	Yes	Presumed de novo or sporadic	LP	[13]
Ma 2021	c.916C>G	Arg306Gly	Basic motif	Missense	N/A	N/A	No	Presumed de novo or sporadic	N/A	[20]
Wang 2022	c.947T>C	Leu316Pro	Leucine zipper region	Missense	Nuclear, zonular, and structural cataract	N/A	No	AD	LP	[21]
Li 2018	c.950A>G	Glu317Gly	Leucine zipper region	Missense	Posterior polar cataract	N/A	No	AD	P	[22]
Hansen 2009	c.958A>G	Lys320Glu	Leucine zipper region	Missense	N/A	Microcornea	No	AD	N/A	[23]

**Table 2 genes-15-00686-t002:** A brief summary of reports of animal and cellular models that demonstrate the relationship between the *MAF* domain and the congenital cataract phenotype [12,14,24,25,26,27,28].

Citation	Model(s) Used	Experimental Finding(s)	Reported Phenotype
[24]	Embryonic and adult mice heterozygous or homozygous for mutated murine *c-maf* gene containing a b-galactosidase (lacZ) gene insertion.	*c-Maf* expression is higher in primary fiber cells than epithelial cells in mice. γ-crystallin expression was not detected in c-Maf-deficient newborns, and αA-, αB-, and β-crystallins were downregulated.	Embryos homozygous for c-maf mutation have abnormal lenses with defective differentiation of lens fiber cells (no elongation).
[25]	*c-maf* knockout mouse model with abnormal lens development	Pax6 and c-Maf mRNAs are expressed in the lens equator.	N/A
[14]	Mouse	A translocation within *Maf*, t(5;16)(p15.3;q23.2), that was isolated from a family with cataract, anterior segment dysgenesis, and microphthalmia, and cloned demonstrated defective lens formation and microphthalmia in mouse embryos.	The null mutant *Maf* mouse embryo demonstrated defective lens formation and microphthalmia.
[26]	COS-1 and human lens epithelial cells (HLECB3) transfected with a reporter (one of three mouse crystallin promoter-luciferase reporters (αA, βB2, and γF) or a MARE-TK-luciferase reporter) and a plasmid encoding *c-MAF, Prox-1*, and/or *Sox-1* in the presence of absence of *CBP* or *p300*.	*c-Maf* expression transactivated each of the promoters. Coexpression of *CBP* or *p300* with *c-MAF* synergistically co-activated each promoter. *c-Maf* likely upregulates crystallin gene expression by recruiting *CBP* and/or *p300* to crystallin promoters.	N/A
[27]	*Ofl* mice with *Maf* mutation (R291Q) demonstrating cataract	The mutation in *Ofl* mice result in selectively altered DNA binding affinities to target oligonucleotides with downstream cascade effects.	Homozygous mice fail to differentiate lens fiber cells (remain columnar epithelium). Heterozygous Ofl mice crossed with different genetic backgrounds generate anterior segment abnormalities.
[28]	Murine c-Maf mutant (ENU424) associated with isolated congenital cataract.	The large Maf transactivation mutation enhances interaction with transcriptional co-activator p300.	ENU424 heterozygotes expressed a mild granular nuclear opacity, and homozygotes expressed a denser and more severe nuclear opacity.
[12]	HEK 293 T cells transfected with pcDNA3.1-MAF expression plasmid and pGL3-crystallin/non-crystallin promoter luciferase plasmid. Control: pRL-TK Renilla luciferase vector	Val271Glu variant (c.812T>A) significantly impaired the transactivation of four crystallin genes (*CRYGA*, *CRYAA*, *CRYBA1*, and *CRYBA4*) involved in lens composition.	Non-syndromic congenital nuclear and lamellar opacities observed in family.

## Data Availability

The results of the genetic testing were included in the manuscript of this paper.

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
