# Peer review of "A Case of Non-Syndromic Congenital Cataracts Caused by a Novel MAF Variant in the C-Terminal DNA-Binding Domain—Case Report and Literature Review"

_genes, 2024, doi:10.3390/genes15060686_

Round 1
Reviewer 1 Report
Comments and Suggestions for Authors
The authors reported a case about the MAF variants and reviewed the C-terminal MAF variants and their associated congenital cataracts and ophthalmic findings. However, several questions should be addressed as follows.
1. Results, 3.1, it would be better to add the family information.
2. Results, 3.1, it would be better to add some figures about the examinations of the patient.
3. Results, 3.2, it would be better to clarify why the authors review the literatures and tell us the main findings from the comparison between the literatures and your results.4. Line 193, ‘We This …’ sentence, maybe ‘We’ should be deleted.
Author Response
- COMMENT: Results, 3.1, it would be better to add the family information.
RESPONSE: Thank you for your comment. We have included the clinical family information in section 2.1 Patient Information and History. No family members had congenital cataracts. In section 3.1, we included that the MAF variant was not identified in either parent.
- COMMENT: Results, 3.1, it would be better to add some figures about the examinations of the patient.
RESPONSE: Unfortunately we do not have photos of this patient’s examinations.
- COMMENT: Results, 3.2, it would be better to clarify why the authors review the literatures and tell us the main findings from the comparison between the literatures and your results.4. Line 193, ‘We This …’ sentence, maybe ‘We’ should be deleted.
RESPONSE: We made changes to Results 3.2 to include an explanation of the reason for literature review and comparison to our patient’s findings.
Reviewer 2 Report
Comments and Suggestions for Authors
Zhao and coworker present a case report showing a novel MAF variant related to non-syndromic congenital cataracts. The manuscript also presents a review of related literature.
Although the paper is a case report, additional information must be provided beyond what has been presented.
1. Experimental data of the mutation obtained through WES analysis in the patient and parents must be presented.
2. The mutation needs to be confirmed by Sanger sequencing in both the patient and parents, accompanied by relevant electropherograms.
3. Figure 2 lacks sufficient information, as does the accompanying caption. Consider adding details such as the protein's size and the amino acid position at the boundaries of each indicated domain. Additionally, including a 3D reconstruction of the protein, both with and without the identified variant, would enhance reader understanding.
4. Was an opinion from the Ethics Committee not obtained? At the very least, the 'Institutional Review Board Statement' should indicate that all procedures conducted in this study were in accordance with the ethical standards outlined in the 1964 Helsinki Declaration and its subsequent amendments or comparable ethical standards.
5. Data Availability Statement: The current statement that 'No new data were created or analyzed in this study' seems incongruent with a scientific article, even if a case report. WES data were indeed obtained and should be presented, along with Sanger sequencing results to validate the observed variant.
Author Response
- COMMENT: Experimental data of the mutation obtained through WES analysis in the patient and parents must be presented.
RESPONSE: We are happy to provide more information. What time of experimental data would you like?
- COMMENT: The mutation needs to be confirmed by Sanger sequencing in both the patient and parents, accompanied by relevant electropherograms.
RESPONSE: The WES was done by a CLIA-certified lab which was added to the manuscript in Results section 3.1.
- COMMENT: Figure 2 lacks sufficient information, as does the accompanying caption. Consider adding details such as the protein's size and the amino acid position at the boundaries of each indicated domain. Additionally, including a 3D reconstruction of the protein, both with and without the identified variant, would enhance reader understanding.
RESPONSE: Thank you for your comment. We have added a 3D reconstruction of the protein.
- COMMENT: Was an opinion from the Ethics Committee not obtained? At the very least, the 'Institutional Review Board Statement' should indicate that all procedures conducted in this study were in accordance with the ethical standards outlined in the 1964 Helsinki Declaration and its subsequent amendments or comparable ethical standards.
RESPONSE: Yes, an IRB approval was obtained and the patient’s parents signed an informed consent. We have provided more detail in Section 2.1
Round 2
Reviewer 2 Report
Comments and Suggestions for Authors
The manuscript did not have any input in the revision to improve the parts I had suggested.
The manuscript remains without a experimental results "shown" in a figure and clearly described in their methodolical procedure. It is not sufficient to state that a certified laboratory has obtained the indicated result; in this case, since the result is the basis of the paper, a co-author belonging to that laboratory should appear. Moreover, a simple electropherogram with Sanger sequencing can be done in a very short time.
Figures 1 and 2 seem rather simple to me... they lack an adequate description (the legends are rather sparse, as are the figures) and as they are shown they add nothing to the results.
No opinion from an ethics committee was indicated, but only that the criteria of the Declaration of Helsinki were followed. I don't know if this is enough for the journal.
Data Availability Statement indicates that "results of the genetic testing were included in the manuscript of this paper." but I don't see results shown, but only a statement of the results obtained in an external laboratory.
Author Response
COMMENT: The manuscript remains without a experimental results "shown" in a figure and clearly described in their methodolical procedure. It is not sufficient to state that a certified laboratory has obtained the indicated result; in this case, since the result is the basis of the paper, a co-author belonging to that laboratory should appear. Moreover, a simple electropherogram with Sanger sequencing can be done in a very short time.
RESPONSE: Thank you for your comment. Our group does not do our own laboratory testing. This testing was done by a CLIA-certified lab that performs clinical genetic testing and provides the testing for our research group. The results from this laboratory can be utilized for clinical decision making and we feel are sufficient for diagnosis for this patient. We have included an additional figure 1B of MAF testing in our patient and parents on IGV.
COMMENT: Figures 1 and 2 seem rather simple to me... they lack an adequate description (the legends are rather sparse, as are the figures) and as they are shown they add nothing to the results.
RESPONSE: Thank you for your comment. We have added additional descriptions and Figure 1B.
COMMENT: No opinion from an ethics committee was indicated, but only that the criteria of the Declaration of Helsinki were followed. I don't know if this is enough for the journal.
RESPONSE: Thank you for your comment. We had incorrectly stated at the end of the paper that the IRB was non applicable although we had put in the body of the manuscript that we had IRB approval. That was an oversight on our part and we appreciate you finding this inconsistently. Our institution's ethics committee (Institutional Review Board) approved our study under protocol IRB 2021-4730 and is now included both in the body of manuscript and in the descriptions at the end. We have provided the journal with a copy of the consent form signed by our patients' parents.
COMMENT: Data Availability Statement indicates that "results of the genetic testing were included in the manuscript of this paper." but I don't see results shown, but only a statement of the results obtained in an external laboratory.
RESPONSE: Thank you for your comment. Our research protocol obtains genetic testing results from an external laboratory. We included Figure 1B as described above.
Round 3
Reviewer 2 Report
Comments and Suggestions for Authors
Authors added the lacking information in the manuscript, as I suggested. Now, I believe, in my opinion, the manuscript in the present form can be accepted for publication in Genes.